# Investigation of the Proteomes of the Truffles *Tuber albidum pico*, *T. aestivum*, *T. indicum*, *T. magnatum*, and *T. melanosporum*

**DOI:** 10.3390/ijms222312999

**Published:** 2021-11-30

**Authors:** Dennis Krösser, Benjamin Dreyer, Bente Siebels, Hannah Voß, Christoph Krisp, Hartmut Schlüter

**Affiliations:** Institute of Clinical Chemistry and Laboratory Medicine, Section Mass Spectrometry and Proteomics, University Medical Center Hamburg-Eppendorf, Martinistraße 52, 20246 Hamburg, Germany; d.kroesser@uke.de (D.K.); b.dreyer@uke.de (B.D.); bente.siebels@gmail.com (B.S.); ha.voss@uke.de (H.V.); christoph.krisp.uke@gmail.com (C.K.)

**Keywords:** truffles, proteomes, bottom-up proteomics, liquid chromatography coupled to mass spectrometry (LC-MSMS), data-independent acquisition (DIA), food fraud

## Abstract

Truffles of the *Tuber* species are known as expensive foods, mainly for their distinct aroma and taste. This high price makes them a profitable target of food fraud, e.g., the misdeclaration of cheaper truffle species as expensive ones. While many studies investigated truffles on the metabolomic level or the volatile organic compounds extruded by them, research at the proteome level as a phenotype determining basis is limited. In this study, a bottom-up proteomic approach based on LC-MS/MS measurements in data-independent acquisition mode was performed to analyze the truffle species *Tuber aestivum, Tuber albidum pico, Tuber indicum, Tuber magnatum,* and *Tuber melanosporum*, and a protein atlas of the investigated species was obtained. The yielded proteomic fingerprints are unique for each of the of the five truffle species and can now be used in case of suspected food fraud. First, a comprehensive spectral library containing 9000 proteins and 50,000 peptides was generated by two-dimensional liquid chromatography coupled to tandem mass spectrometry (2D-LC-MS/MS). Then, samples of the truffle species were analyzed in data-independent acquisition (DIA) proteomics mode yielding 2715 quantified proteins present in all truffle samples. Individual species were clearly distinguishable by principal component analysis (PCA). Quantitative proteome fingerprints were generated from 2066 ANOVA significant proteins, and side-by-side comparisons of truffles were done by T-tests. A further aim of this study was the annotation of functions for the identified proteins. For *Tuber magnatum* and *Tuber melanosporum* conclusive links to their superior aroma were found by enrichment of proteins responsible for sulfur-metabolic processes in comparison with other truffles. The obtained data in this study may serve as a reference library for food analysis laboratories in the future to tackle food fraud by misdeclaration of truffles. Further identified proteins with their corresponding abundance values in the different truffle species may serve as potential protein markers in the establishment of targeted analysis methods. Lastly, the obtained data may serve in the future as a basis for deciphering the biochemistry of truffles more deeply as well, when protein databases of the different truffle species will be more complete.

## 1. Introduction

Truffles are ascomycete fungi in the genus *Tuber* that form subterranean fruiting bodies. They often grow in symbiotic relationships with the roots of trees and have a pronounced taste and, more important, smell to attract animals for spreading their spores while being dug up and eaten [1]. Because of these organoleptic properties, some truffles, especially the French black Perigord truffle (*Tuber melanosporum*) and Italian white piedmont truffle (*Tuber magnatum*), are highly prized foods with prices over 3000 €/kg [2]. Other mostly European truffle species, such as the summer truffle (*Tuber aestivum*), are more widespread and therefore less expensive. Because of their distinct aroma, many studies on truffles focused on the identification of mostly volatile compounds [3,4,5,6,7]. Additionally, as an expensive food, truffles are a worthwhile target for food fraud [8]. Food fraud is not yet clearly defined by the commission of the European Union, and as a result, the food industry is susceptible to food fraud. Robson et al. discussed the different definitions in detail [9]. In the case of truffles, product mislabeling is common. Less aromatic and thereby cheaper truffle species are often declared and sold as the expensive, very tasty species. Recently, Karkouri et al. developed a MALDI-TOF based method for the fast classification of truffle samples by analyzing intact proteins [10]. In 2010, black truffle *Tuber melanosporum* was the first *Tuber* species with a fully sequenced genome [11]. Recent studies on whole truffle proteomes are very scarce. In the past, mostly gel-based approaches for tackling the truffle proteomes and comparing electrophoretic patterns were used [12,13]. In 2013, Islam et al. were the first to do a combined deep proteomics and bioinformatics approach for taking a comprehensive look on a whole global truffle proteome, using the black Perigord truffle (*Tuber melanosporum*, strain Mel28) as an example. They functionally annotated the proteome and used sequential BLAST search strategies to match proteins to fungal homologues. For verification, they did a gel-based bottom-up proteomics approach. By this approach, 836 proteins were identified and taken into account for annotation [14]. In their work they annotated the functions of 20% of all potential black truffle proteins and had verifications for 846 of them. Today, the number of functional annotations has increased with sequencing of *Tuber magnatum* and *Tuber aestivum* in 2018 [15], as well as *Tuber borchii* [16]. Additionally, within the last years, the proteomic field highly benefited from increasing performance in instrumentation and software. Data-independent acquisition (DIA) was introduced as a powerful tool for bottom-up proteomics, possessing high identification rates and low missing values, enabling a better quantification [17,18]. Because DIA-MS2 spectra are highly complex, commonly a spectral library is needed for data extraction. Spectral libraries are usually generated by running a pooled part of the samples of interest in data-dependent acquisition (DDA) mode.

The two-dimensional chromatographic fractionation separates peptides according to their chemical properties and limits the number of molecules present in each MS scan. For an individual MS scan only a limited number of peptides can be transferred to the MS2 level and thus be identified. Therefore, the prefractionation of a pooled sample—for example, by high pH reversed phase chromatography—into multiple, individually measured fractions increases the peptide identification rate [19,20]. Here, we performed a comprehensive comparison of different truffles of commercial interest on the proteome level by using state-of-the-art bottom-up proteomics-based LC-MS/MS. We generated quantitative protein abundance fingerprints for five truffle species and worked out the differences between species by species-to-species comparisons. All obtained data, deposited in Pride (PXD027871), can be used as a repository for future research on truffles, which are still limited in their characterization. Furthermore, the data will aid in clarifying the identity of truffles species in case of food fraud.

## 2. Results

### 2.1. Testing Sample Preparation for LC-MS/MS Based Bottom-Up Proteomics

To determine which methods recovered the most proteins for ground truffle powder, three different established sample preparation protocols for bottom-up proteomics by LC-MS/MS were tested to prepare ground truffle powder from one sample of *T. indicum*. One protocol used sodium deoxycholate (SDC) for lysis and protein extraction, followed directly by tryptic digestion in solution. The other protocols either used sodium dodecylsulfate (SDS) or urea for lysis and protein extraction, followed by the filter aided sample preparation (FASP) approach, utilizing a centrifuge spin filter with a 30 kDa molecular cutoff. First, the effectiveness of protein extraction was studied with SDC (yield according to BCA test 3.6 ± 0.3 µg/µL), urea (4.5 ± 0.3 µg/µL), or SDS (6.2 ± 0.2 µg/µL). For a comparison of the different protein extraction approaches, the same amounts of extracted protein were tryptically digested and generated peptides measured by LC-MS/MS in DDA mode. The protein identification rates are plotted in Figure 1.

Using the SDC protocol, 1147 ± 8 proteins were identified. The FASP approach for preparing SDS extracted samples resulted in 1205 ± 11 proteins. The same approach for the urea extracted samples led to the identification of 630 ± 41 proteins. Therefore, the FASP approach using SDS extracted samples worked best for the preparation of truffle powder regarding a high and reproducible identification rate, followed by the SDC approach.

Next, the quantitative reproducibility of sample preparation protocols with SDC and SDS extracted proteins was assessed. For this, only the 1042 proteins with quantitative values in all samples were considered. Coefficients of variation (CV) for protein areas were calculated and compared. Mean CV for proteins identified by SDC sample preparation was 11.0%, and for SDS extracted proteins processed with the FASP approach 12.9%. For following quantitative, non-targeted, and label-free LC-MS/MS approach in DIA mode to generate proteomics profiles of different truffle species, the SDC sample preparation was chosen. The protocol had a high extraction efficiency, a high and reproducible protein identification rate, and had the least sample handling steps.

### 2.2. Spectral Library Generation for DIA Measurements

After finding a suitable bottom-up proteomics sample preparation protocol, a spectral library from available samples was created for subsequent measurements in DIA mode. Samples for each *T. aestivum*, *T. magnatum,* and *T. melanosporum* were pooled and fractionated by a high pH reversed phase approach on HPLC before measurement in DDA mode on a low pH LC-MS/MS system. Each species was searched separately against their corresponding protein sequence database from UniProt. For *T. aestivum* 3095 proteins and 18,234 peptides were identified, for *T. magnatum* 3544 proteins and 24,564 peptides, and for *T. melanosporum* 3519 proteins and 26,686 peptides. Results were combined for a spectral library containing 9170 proteins and 51,628 peptides. The generated spectral library was used for the quantification of all truffle species analyzed, including *A. Pico* and *T. indicum.* Library generation was not performed for these species, as no (*A. Pico*) or insufficient (54 unreviewed proteins; *T. indicum*) FASTA information was applicable due to incomplete genome sequencing of these species.

### 2.3. DIA Measurements of Different Truffles

Measurements in DIA mode were done for all 72 samples present (28 samples of *T. aestivum*, 4 samples of *T. Albidum Pico*, 10 samples of *T. indicum*, 19 samples of *T. magnatum,* and 11 samples of *T. melanosporum*).

Principal component analysis (PCA) was performed with abundance values of 2715 proteins identified in all samples. Figure 2 shows the sample projection for the two first principal components.

Projection of the two first principal components revealed a high similarity and clustering for samples of *T. aestivum*, *T. albidum Pico*, *T. magnatum*, *T. melanosporum,* and *T. indicum*. *T. aestivum* and *T. magnatum* were the most distinguishable species in the projection. Still being separated in the projection, *T. albidum Pico*, *T. melanosporum,* and *T. indicum* clustered together more closely. Separation from them was driven by the second component and was not strongly impacted by the first component. *T. indicum* and *T. melanosporum* were the species that clustered most closely in the depicted projection.

To continue the global comparison of truffle species, ANOVA testing (analysis of variance) was done. This multi-testing approach was used to identify significant differences in the abundance values for proteins between different truffle species. A total of 2066 proteins were ANOVA significant out of the 2715 proteins identified in all samples. The obtained heat map displaying significant proteins after ANOVA testing is depicted in Figure 3.

The dendrogram showed that all samples within the same species were very similar and clustered together, whereas different species were separated after hierarchical clustering analysis. This was expected because a separation was already visible in the principal component analysis sample projection plot. While still segregated, *T. melanosporum* and *T. indicum* clustered together most closely and with a small distance to *T. albidum Pico. T. aestivum* and *T. magnatum* were the most distant from each other and from the other species, which is consistent with previous findings.

For getting a better insight in differences between species, the 2715 proteins identified in all samples were used to perform species-against-species comparisons by Student’s T-test, and significant regulated proteins (1% FDR) were filtered for an at least two-fold change. Results of this species-to-species comparison are listed in Figure 4.

Each of the five species is assigned a color. The species-versus-species comparisons take place horizontally versus vertically. For each species-versus-species comparison, the number of significantly upregulated proteins in both species within that comparison is shown. Thus, there are two numbers of upregulated proteins for each comparison. The number of upregulated proteins is also color coded to match the corresponding species in the comparison. For example, in the comparison of *T. aestivum* (color coded green) with *T. magnatum* (color coded orange) there are 389 proteins present in *T. aestivum* with an at least two-fold higher abundance than in *T. magnatum.* Correspondingly, there are 483 protein present with an at least two-fold higher abundance in *T. magnatum* compared to *T. aestivum*. Strongest differences in abundance of proteins were observed between *T. melanosporum* and *T. magnatum*. In total, 1174 proteins were significantly different in abundance with an at least two-fold change. Of these 1174 proteins significantly different in abundance, 589 proteins had an at least two-fold higher abundance in *T. magnatum,* while 585 proteins were more abundant in *T. melanosporum*. Using the same criteria, the second largest differences were 1044 proteins between *T. magnatum* and *T. aestivum* and 964 proteins between *T. melanosporum* and *T. aestivum*. The smallest differences were observed between *T. indicum* and *T. albidum Pico*, with only 242 proteins significantly different in abundance and a two-fold change.

### 2.4. Gene Ontology Enrichment Analysis

To estimate the biological processes that predominate in the various truffle species, ANOVA significant proteins were first functionally annotated. A gene ontology enrichment analysis was performed on a previously obtained list of ANOVA significant proteins. From the 2066 ANOVA significant proteins, 1325 had functional annotations and were linked to biological processes. Table 1 shows a list of the ten most enriched biological processes. The greatest enrichment was observed for various metabolic and oxidation–reduction processes.

From the comparisons, species-to-species lists of differentially regulated proteins were obtained. Analysis of gene ontology enrichment was done to gain insight into the underlying functional annotation of the regulated proteins. Annotation of upregulated proteins was done according to their annotated biological processes. However, on average, only 63% of the proteins on the lists of identified and differentially regulated proteins had an annotation for biological processes. Most often, enrichment for metabolic processes and oxidation reduction processes occurred in the comparisons. Metabolic processes included processes involving small molecules, organic acids, carboxylic metabolic processes, and processes involving oxoacids. Further, an enrichment for organonitrogen compound metabolic processes was found in many of the species comparisons. The 15 most enriched processes for proteins upregulated in *T. magnatum* when compared to *T. indicum* are shown exemplarily in Table 2.

Interestingly, *T. magnatum* was found to be enriched for sulfur-compound metabolic processes (sulfur amino acid and cysteine metabolic processes, sulfur compound metabolic process, and hydrogen sulfide metabolic process) when compared to *T. indicum*. In each comparison, *T. magnatum* showed higher enrichment than other truffles for at least one of these sulfur metabolic processes. Sulfur metabolism processes were also discovered to be top hits for *T. melanosporum* when compared to other truffles. When *T. magnatum* and *T. melanosporum* were compared, these previously found sulfur-related processes did not rank among the top hits in the lists of enriched processes for either of the species.

## 3. Discussion

Typical workflows for extraction of proteins and tryptic proteolysis by bottom-up proteomics were tested on a sample of ground truffle powder. Buffers containing sodium deoxycholate (SDC), sodium dodecyl sulfate (SDS), and urea were tested for protein extraction. Afterwards, an estimation of protein concentration was done by BCA assay. Most effective disruption of membranes and solubilization of proteins was achieved with SDS, indicated by the highest extraction efficiency (6.2 µg/µL). The efficiency of extraction with urea (4.5 µg/µL) was superior to that of buffer with SDC (3.6 µg/µL). Nevertheless, with all extraction buffers, a sufficient protein amount was obtained to continue processing of samples with no limitation regarding available material. No additional extraction step was necessary to break down the truffle cell wall, which contains chitin [22].

High pH reversed phase chromatography and a concatenated pooling scheme are widely used in proteomics [19,20,23]. Applying these approaches on truffles, a comprehensive spectral library with the highest proteome coverage of truffles up to now was created. Due to the scarcity of truffle protein databases on UniProt and the large differences in sample numbers between truffles, only samples of *T. magnatum*, *T. aestivum*, and *T. melanosporum* were used for the library. Missing out on proteins specific for both *T. albidum Pico* and *T. indicum*, which were not used for generating the spectral library, is likely possible. In this case, DIA provides the option to later expand spectral libraries with more samples and new databases. Datasets can then be re-analyzed without having to re-measure every sample. Since 2715 proteins could nevertheless be identified in all samples present, including those of *T. albidum Pico* and *T. indium*, by comparison with the generated spectral library, it was found to be suitable for analysis of all five truffle species.

Quantitative values of proteins were used in principal component analysis and hierarchical clustering. In both, different truffle species were exceptionally well separated. In protein fingerprints generated by ANOVA testing, this high degree of difference between truffles is also reflected. ANOVA significant proteins were found in 2066 of the 2715 proteins identified in all samples. Therefore, truffles differ highly on the proteomic level. Identified protein marker candidates with different abundance over species and obtained species specific fingerprints could be used in the food industry as a starting point for authentication of truffle samples. Selected-reaction monitoring (SRM) or enzyme-linked immunosorbent assays (ELISA) as quicker and targeted quantitation methods can be established for the chosen truffle proteins and can lead to identification of unknown truffle samples.

After determining the proteomic differences for all truffle species, the availability of information about truffles became a bottleneck for biological interpretation. Only 14 of the 7491 protein entries from *T. melanosporum* in the UniProt database have been reviewed. *T. magnatum* (9412 entries) and *T. aestivum* (9311 entries) are both completely unreviewed. Only 22 of 39,055 entries, corresponding to 0.72%, of all *Tuber* genus proteins deposited on UniProt are reviewed, as current of 4 December 2020. Furthermore, many proteins are uncharacterized, and their functions are not annotated or inferred from their similarity to other proteins. Only 57.6% of all protein entries in *T. magnatum* are functionally annotated, 64.7% in *T. melanosporum* and 46.8% in *T. aestivum*. Nonetheless, an examination of the underlying biological processes revealed some conclusive findings based on the obtained proteomic data.

When the white truffle *T. magnatum* was compared to the black Chinese truffle *T. indicum*, sulfur amino acid and cysteine metabolic processes, sulfur compound metabolic processes, and hydrogen-sulfide metabolic processes were found to be over-represented in *T. magnatum* up-regulated proteins. In truffles, different volatile organic compounds determine taste and the economic value of a species. These include several aldehydes, alcohols, esters, ketones, terpens, and sulfur-containing compounds [3,24]. Sulfur-containing volatile organic components (VOC) play an important role for truffle aroma [4], especially for *T. magnatum* [3,24]. These differences in VOC level are therefore in accordance with our findings on proteomic level, where an enrichment of the GO term sulfur compound metabolic process was evident among ANOVA significant proteins. This includes the enrichment of the GO term “Thioester Metabolic processes” (Appendix A). A higher abundance of proteins responsible for processes of sulfur-metabolism can lead to more sulfur-containing VOCs in *T. magnatum*, resulting in the stronger aroma. In comparison with the other truffles *T. indicum*, *T. aestivum,* and *T. albidum*, sulfur-metabolic processes were over-represented again in T. magnatum. Similar results were obtained for the black truffle *T. melanosporum*, also known for a strong aroma and sulfur-containing VOCs [5].

Comparing all the different truffles side-by-side, many different metabolic processes and oxidation–reduction processes were in the lists of over-represented biological processes. In addition, different *Tuber* species could be clearly distinguished on the basis of proteins, associated with the GO term “Alcohol Metabolic Processes”. In particular, the proteins involved in the myo-inositol biosynthesis (inositol-3-phosphate synthase and inositol-1-monophosphatase) were found to be more abundant in *Tuber* Melanosporum and *Tuber* albidum pico, compared to all other species. They can be possibly linked to different growth conditions for each species. The VOC composition and the associated taste of truffles is determined by multiple factors, such as genetically differentiated populations [25], different bacterial [26] or fungal [27,28] communities associated with the fruiting body, and most strongly by geographical area of origin [3], connected to the climate. Exogenous factors are important in the composition of the metabolomes in plant metabolomics. Warmth, heat waves, drought, or frost as well as soil composition, elevation, sun exposure, and other factors, must all be considered. All of these variables can have a significant impact on a plant’s metabolome. Seasonal differences in plants grown in the same geographical location can have a significant impact on metabolomic levels [29]. The same observations will most likely apply to truffles.

Proteins, through their enzymatic activity, are responsible for all metabolic reactions. As a result, the observed enrichment of metabolic and Redox processes in proteins for various truffles is consistent with previous metabolomics findings. Proteins, through their enzymatic activity, are responsible for all metabolic reactions. As a result, the observed enrichment of metabolic and Redox processes in proteins for various truffles is consistent with previous metabolomics findings. The proteomic differences between different truffles is predominantly driven by their genetic similarity and phylogenetic relation. Interestingly, the grade of similarity among the proteomic profiles of different truffle species reflects their phylogenetic relationship, rather than their soil of origin. Gene sequencing [30] as well as an integrated phylogenetic analysis, using internal transcriber spacer (ITS) sequencing and MALDI-TOF protein data [31], describe the closest evolutionary relationship between *T. indicum* and *T. melanosporum.* In proteomic data, the highest similarity between the protein profiles of these species can be observed, based on hierarchical clustering (Figure 3) and the first PC in PCA, accounting for 29.9% of the explained variance. According to the literature, *T. indicum and T. melanosporum* can be clearly separated from *T. magnatum* and *T. avestivum,* who are more closely related. In the current study, this is reflected by Component 2 in PCA accounting for 19.6% of the explained variance. Based on these data, for *T. albidum pico*, a close evolutionary relation to *T. indicum* and *T. melanosporum* can be suspected according to PC1. However, to our current knowledge, there is no study investigating the phylogenetic relationship of *T. albidum pico*, which represents the least studied truffle species. In 2018, Vahdatzadeh et al. suggested that the aroma variability among different genetic *Tuber* species is predominately linked to amino acid catabolism through the Ehrlich pathway [30]. Here, we found a significant enrichment of the GO term “amino acid catabolic processed” among the significantly changed proteins between different truffle species. However, the enrichment was predominately driven by the abundance distribution of proteins, associated with the glycine cleavage system (Uni Prot ID: D5GPW1, A0A292Q6M5, A0A317SZS6, and A0A292PJW0)—responsible for the conversion of glycine residues to 5,10-methylene-H_4_ folate—which shows an increased abundance in *Tuber* aestivum.

The impact of cysteine degradation products on the aromatic composition of truffles has not been studied yet. Moreover, volatile degradation products of leucine, isoleucine, phenylalanine, and methionine have been described in this context. The Ehrlich pathway deamidates amino acids to alpha-a-keto acids that are decarboxylated to aldehydes, which can be reduced to alcohols [30]. To our current knowledge, the exact enzymes responsible for the generation of these volatiles have not been disclosed for the analyzed *Tuber* species. Due to the lack of information, only incomplete FASTA protein databases can be provided. A FASTA database, containing all theoretical protein sequences for a species, is required for the identification of proteins from bottom-up LC-MS/MS data. Therefore, it is highly recommended to reanalyze the raw data, provided in this study, via PRIDE as soon as complete information on the truffle proteome is available to get deeper insights into proteomic differences between *Tuber* species that can potentially explain differences in VOC signatures and taste.

## 4. Materials and Methods

### 4.1. Chemicals

Sodium deoxycholate (SDC), urea, sodium dodecyl sulfate (SDS), dithiothreitol (DTT), iodocetamide (IAA), ammonium bicarbonate (ABC), methanol, and formic acid (FA) were purchased from Sigma-Aldrich (St. Louis, MO, USA). Triethylammonium bicarbonate (TEAB) and the Pierce Protein Assay Kit were purchased from Thermo-Fisher (Waltham, MA, USA). Sequencing grade modified trypsin was purchased from Promega (Madison, WI, USA). LC-MS grade water and LC-MS grade acetonitrile (ACN) were purchased from Merck (Darmstadt, Germany).

### 4.2. Truffle Samples

All truffle powder samples were obtained from Prof. Markus Fischer’s group at the University of Hamburg in collaboration with the Trüffelkontor GmbH (Waldmünchen, Germany) and the La Bilancia Trüffelhandels GmbH (Munich, Germany). After washing with ultrapure water, truffles were stored at −80 °C. Truffles then were homogenized using a GM 300 knife mill (Retsch, Haan, Germany) and one part truffle with one part dry ice. Afterwards, the powder was lyophilized for 48 h and then stored at −80 °C.

### 4.3. Protein Extraction from Truffle Powder

Samples of truffle powder were mixed in triplicate with one of three following buffers for protein extraction: 1% SDC/ 100 mM TEAB, abbreviated as SDC buffer; 8 M urea/50 mM ABC, abbreviated as urea buffer; or 5% SDS/50 mM ABC, abbreviated as SDS buffer. The buffer-to-truffle powder ratio was 14 µL of buffer to 1 mg of powder. Protein extraction with either SDC or SDS buffer was done by boiling samples for 10 min at 99 °C, followed by sonication with a probe (Bandelin Sonoplus, 30% energy for 30 s). Extraction with urea buffer was done by incubation on ice for 30 min before samples were sonicated with a probe (Bandelin Sonoplus, 30% energy for 30 s). Samples were centrifuged at 10,000× *g* for 10 min before the obtained supernatant was transferred to a new reaction tube. An estimation of protein concentration was done for the supernatant by using the Pierce Protein Assay Kit according to the manufacturer’s instructions.

### 4.4. Tryptic Digestion of Extracted Proteins

#### 4.4.1. Tryptic In-Solution Digestion of Proteins Extracted with Sodium Deoxycholate

A total of 20 µg of protein extracted with SDC buffer was diluted to a total volume of 100 µL with SDC buffer. Disulfide bonds were reduced by adding 1 µL of 1 M DTT to a final concentration of 10 mM. Samples were incubated in a heating block for 30 min at 56 °C. Reduced cysteines were blocked by adding 4 µL of 0.5 M IAA to a final concentration of 20 mM and incubation in the dark for 30 min at 37 °C. In order to obtain a ratio of 1 part trypsin to 100 parts protein, 0.2 µg of trypsin was added. Proteolysis by trypsin was performed at 37 °C overnight. Tryptic activity was stopped by adding FA to a final concentration of 2%. SDC was precipitated by centrifugation of samples at 16,000× *g* for 5 min. The obtained supernatant was then transferred to a new reaction tube. Prior to measurements by LC-MS/MS, supernatants were dried by using a vacuum centrifuge and resuspended in 0.1% FA with 1 µg/µL concentration.

#### 4.4.2. Filter Aided Sample Preparation (FASP) for Urea and Sodium Dodecyl Sulfate Extracted Proteins

Proteins extracted with buffers containing urea or SDS were processed with the FASP method [31]. A total of 20 µg of extracted protein was transferred into a centrifugal spin filter device with a membrane enabling a retention by molecular weight with a cut-off of 30 kDa (Amicon Ultra 0.5 mL, Merck Millipore (Billerica, MA, USA)). A total of 200 µL buffer containing 8 M urea in 50 mM ABC, abbreviated as UA solution, was then added, and the centrifugal spin filter devices were placed in a centrifuge at 14,000× *g* for 10 min. The procedure was performed a second time. Reduction was carried out by adding 50 µL of 10 mM DTT and incubating at 56 °C for 30 min, followed by centrifugation at 14,000× *g* for 5 min. Alkylation was done by adding 50 µL of 20 mM IAA, incubating in the dark at 37 °C for 30 min, again followed by centrifugation at 14,000× *g* for 5 min. Afterwards, samples were washed two times by addition of 100 µL UA solution and centrifugation at 14,000× *g* for 10 min. This step was followed by two more washes by addition of 200 µL 50 mM ABC buffer and centrifugation at 14,000× *g* for 10 min. Proteolytic cleavage was performed by trypsin in a 1:100 ratio and incubation overnight at 37 °C. Tryptic peptides were obtained by centrifugation of the spin filter device at 14,000× *g* for 10 min. Collected peptides were dried in a vacuum centrifuge and resuspended with 0.1% FA to a concentration of 1 µg/µL prior to LC-MS/MS.

### 4.5. LC-MS/MS Parameters and Data Processing for Testing Sample Preparation

Measurements by LC-MS/MS were done on a quadrupole-orbitrap mass spectrometer of the Q Exactive series from Thermo Fisher coupled to a UPLC system of the nanoAcquity series from Waters. For each analysis, autosampler injection of 1 µg peptides was done onto a reversed phase trapping column (Acquity UPLC Symmetry C18; pore size 100 Å, particle diameter 5 µm, 180 µm inner diameter and 20 mm length) and separated on a reversed phase separation column (Acquity UPLC Peptide BEH C18; pore size 130 Å, particle diameter 1.7 µm, 75 µm inner diameter and 200 mm length). Trapping was done with a flow rate of 15 µL/min for 5 min and with 99% solvent A (0.1% FA in water) and 1% solvent B (0.1% FA in ACN). Peptides were separated and eluted with a linear gradient from 1 to 30% solvent B over 60 min. The eluting peptides were infused in a quadrupole-orbitrap mass spectrometer (Q Exactive). MS1 scans were performed in positive mode over a scan range of 400–1200 *m/z*. The orbitrap resolution was set to 70,000 with an AGC target of 1E6 and a maximum injection time of 240 ms. Peptides with the charge states 2–5 over the intensity threshold of 100,000 were isolated with a 2 *m/z* isolation window in Top15 mode and fragmented with a normalized collision energy of 28%. The fragments were measured with an orbitrap resolution of 17,500, AGC target of 10^5^ and 50 ms maximum injection time. Already fragmented peptides were excluded the next 20 s.

Generated raw files were then loaded into the MaxQuant (Version 1.6.2.10) software and processed as individual experiments. In silico generated peptides were digested by trypsin with a maximum of two allowed missed cleavages, a peptide length of 6 or more amino acids, and 6000 Da as maximal peptide mass. Variable modifications were methionine oxidation, a conversion of glutamine to pyro-glutamic acid, and the acetylation of protein N-termini, while the only fixed modification was set for cysteine carbamidomethylation. First, a precursor search was performed with an error tolerance of 20 ppm and the main search with 4.5 ppm, while the error tolerance for fragment spectra was set to 20 ppm. The Perigord Black Truffle database (Strain Mel28) was used for identification, which was downloaded from UniProt on 13 July 2017 with 4113 entries. The rate of false discovery for both proteins and peptide spectrum matches was set to 1%. Quantification of proteins was done with the second peptides, and match between runs function was enabled, considering all identified razor and unique peptides.

### 4.6. Quantitative Analysis of Different Truffle Species

Obtained truffle samples of different species were prepared according to the SDC protocol, described above.

### 4.7. High pH Reversed Phase Fractionation

For high pH fractionation, pooled samples of the species *T. aestivum*, *T. magnatum*, and *T. melanosporum* were used. For each of these three species, 5 samples with the same protein amount were pooled. For this, the samples were resolved in 10 mM ammonium hydrogen carbonate and adjusted to pH 8. Then, 15 µg was taken from each sample of a species and pooled together. From the peptide pool of each species, 50 µg was used for high pH fractionation on HPLC of the Agilent 1200er series, which was connected to an Äkta Prime fractionator. A polymer-based monolithic reversed phase column with 1 mm inner diameter and 250 mm length was used for peptide separation (Thermo ProSwift RP-4H) at a flow rate of 200 µL/min. Solvent A consisted of 10 mM ammonium hydrogen carbonate and solvent B of 80% ACN/10 mM ammonium hydrogen carbonate, with both solvents checked for a pH value of around 8. Over the first 29 min of the high pH reversed phase run, each minute a fraction was collected, consisting of 200 µL. The first 9 collected fractions, corresponding to the first 9 min, were subsequently pooled into three fractions. The remaining fractions were then pooled by a concatenated scheme. Minute 10 was pooled with minute 20, minute 11 with minute 21 until minute 19 and minute 29, resulting in 13 fractions in total. Pooled fractions were then acidified with formic acid to an end concentration of 1% (v/v), dried in a speedvac, and resolved in 0.1% FA before LC-MS/MS analysis.

### 4.8. Parameters for LC-MS/MS Measurements of Different Truffles in DDA and DIA Mode

Data-dependent acquisition (DDA) runs for spectral library generation and data-independent acquisition (DIA) runs were done with a quadrupole-orbitrap-iontrap mass spectrometer of the Orbitrap Fusion series from Thermo Fisher, which was connected to a UPLC system of the Dionex Ultimate 3000 series. For each analysis, 1 µg of tryptic peptides was loaded by the autosampler onto the reversed phase trapping column (Acquity UPLC Symmetry C18; pore size of 100 Å, particle diameter of 5 µm, 180 µm inner diameter and 20 mm length) followed by separation on a subsequent reversed phase column (Acquity UPLC Peptide BEH C18; pore size of 130 Å, particle diameter of 1.7 µm, 75 µm inner diameter and 200 mm length). Peptide trapping was done at a flow rate of 10 µL/min for 5 min with 99% solvent A (0.1% FA in water) and 1% solvent B (0.1% FA in ACN). Separation and subsequent elution of peptides was performed with a linear gradient from 1 to 30% solvent B over 60 min. For washing, there was an increase in solvent B to 95%, which was held for 2 min before decrease to 1% B in 1 min and equilibration at this solvent composition for 15 min.

For spectral library generation, peptides eluting from the column were infused into a quadrupole-orbitrap-iontrap mass spectrometer of the Orbitrap Fusion series from Thermo Fisher. Measurements were done in data-dependent acquisition mode. MS1 scans were performed in positive mode over a scan range of 390–1010 *m/z* with orbitrap detection at a resolution of 120,000, an AGC target of 2 × 10^5^, and maximum injection time of 120 ms. Peptides with the charge states 2–5 over the intensity threshold of 10,000 (with precursor priority to highest intensity) were isolated by the quadrupole with a 1.6 *m/z* isolation window in TopSpeed mode with 3 s cycle time and fragmented with a normalized collision energy of 30%. Fragments were measured with an orbitrap resolution of 30,000, AGC target of 1 × 10^5^, and 50 ms maximum injection time. Already fragmented peptides were excluded the next 15 s.

DIA measurements were done on the same instruments with the same LC parameters. Thirty DIA windows of 20 *m/z* covering the scan range of 400–1000 *m/z* were set in Skyline (Version 20.1) and adjusted by the function optimize window placement. After 15 MS2 scans, a MS1 scan was performed. Parameters for the positive mode MS1 scan were a scan range from 390 to 1010 *m/z*, Orbitrap resolution of 60,000, AGC target 2 × 10^5^, and maximum injection time of 50 ms. MS2 scans in positive mode were done with quadrupole isolation of a 20 *m/z* isolation window, Orbitrap resolution of 30,000, AGC target of 1 × 10^5^, maximum injection time of 50 s, and a normalized collision energy of 28%.

### 4.9. Data Processing Steps for Generation of the Spectral Library

Generated raw files were loaded into the ProteomeDiscoverer (Version 2.0) software, where all three species were analyzed as individual experiments. Fractions collected by high pH fractionation for each species were also processed as fractions and consisted of 13 raw files. Tryptic digestion was performed in silico with a maximum of two allowed missed cleavages and a defined peptide length between 6 and 144 amino acids. Variable modifications were set to methionine oxidation and conversion of glutamine to pyro-glutamic acid at the N-terminus on the peptide level. For protein level, the loss of starter methionine, acetylation on N-termini, and the combination of both was allowed with fixed modification being only set for cysteine carbamidomethylation. For the precursor search the error tolerance was set to 10 ppm, while fragment spectra had to be matched with a 20 ppm tolerance. For protein identification, the corresponding databases for the three Tuber species from UniProt were used. For Tuber magnatum, the TrEMBL database with 9412 entries was used. The protein database for Tuber aestivum was the TrEMBL database with all 9311 entries and for Tuber melanosporum the SwissProt and TrEMBL database with 7494 entries. All databases were downloaded on 6 April 2020. The generated result files of ProteomeDiscoverer from the individual searches for the three species were then merged into a single result file combining all identified proteins.

### 4.10. Data Processing of DIA Measurements

For spectral library generation and processing of data from DIA measurements, Skyline (version 20.1) was used [32]. To generate the spectral library, the merged result file of the high pH fractionated truffle samples from ProteomeDiscoverer was loaded into Skyline together with a combined background database in fasta format.

Transition settings were defined as precursor charges of 2, 3, 4, and 5, and product ions were set to 5 with a minimum of 4, consisting of precursor, b-ions, and y-ions. Extraction of windows was performed according to the defined parameters in the measurement method, while matching of chromatograms was done within a 5 min window. The removal of repeated peptides was done prior to export and filtering for a dotp value in the best sample of 0.85 or higher. The exported result file from Skyline contained the total area fragments of peptides, which were consolidated into values for corresponding proteins. The file with consolidated protein values was used for statistical analysis in Perseus (Version 1.6.2.3). The quantitative values for fragment area after consolidation were loaded as main columns. After log2 transformation, quantitative protein values were normalized by subtraction of the median area for proteins. Then, ANOVA testing, Student’s T-test, analysis for hierarchical clustering, and principal component analysis (PCA) were done. For ANOVA testing and Student’s T-test, comparisons were done with use of a corrected *p*-value (permutation based). The used significance threshold with a *p*-value of 5% or 1% is indicated in the Results section

### 4.11. Functional Annotation

Annotation files for *T. melanosporum* and *T. magnatum* were downloaded from the website for Gene Onotolgy Annotation [33]. Protein sequences for *T. aestivum* were obtained from the UniProt database and consisted of 9312 TrEMBL entries. Annotation was performed with a tool for annotation from the Gene Ontology Annotation website [34]. Then, for the three truffle species a combined file for functional annotation was created, and a further file in obo 1.2 format for gene ontology was downloaded from the Gene Ontology website [35]. Annotation of protein function was then performed with BiNGO [36] version 3.0.4, which was used as a plugin tool for the platform Cytoscape [37], version 3.8.0. The parameters used for finding over-represented protein categories after correction for biological processes were hypergeometric testing with Benjamini and Hochberg false discovery rate correction at a significance level of 0.05.

## 5. Conclusions

In this study, we successfully applied a data-independent acquisition (DIA) bottom-up proteomics approach following the hypothesis that truffle species can be differentiated and identified by the proteome profiles. The DIA approach included the generation of a comprehensive spectral library consisting of 9170 proteins from *T. magnatum*, *T. melanosporum,* and *T. aestivum*. Species-specific proteome profiles of *T. magnatum*, *T. melanosporum*, *T. aestivum*, *T. indicum,* and *T. albidum* were obtained with DIA-based differential quantitative proteomics. The reproducible profiles are very well separated from each other as demonstrated by PCA and hierarchical clustering. Using ANOVA testing to select differentially abundant proteins between different *Tuber* species, individual proteomic fingerprints were generated.

With ANOVA testing, fingerprints for the different species were generated. Species were compared side-by-side, and proteins of different abundance were analyzed for enriched biological processes. Truffles differ most in proteins responsible for various metabolic and redox processes. Further conclusive links to a stronger aroma of *T. magnatum* and *T. melanosporum* were found with enrichment of proteins responsible for sulfur-metabolic processes. We suggest using our results and data, which are present in the data base PRIDE, for further future investigation of the biochemistry of truffles. For food analysis laboratories the profiles may serve for authentication of truffle species.

## Figures and Tables

**Figure 1 ijms-22-12999-f001:**
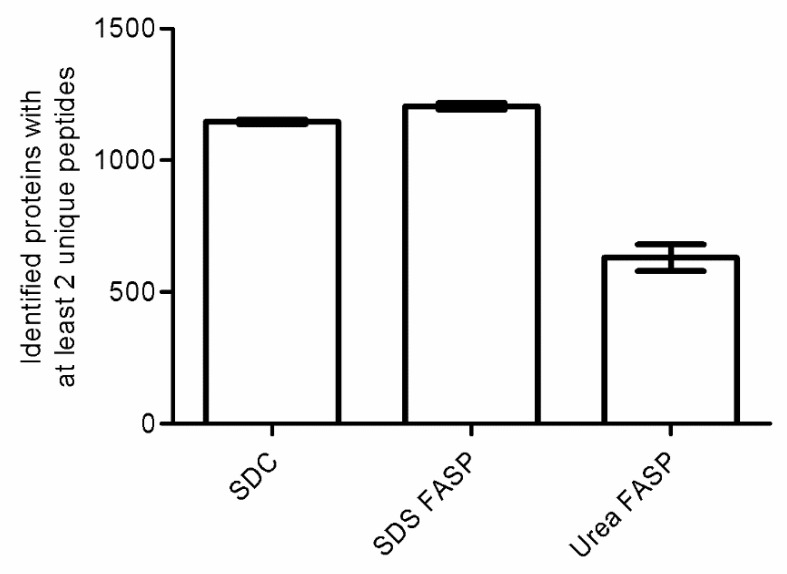
Bar plot comparing the number of identified proteins using different protocols for protein extraction and tryptic digestion from truffle powder of *T. indicum*. Proteins had to be identified with at least two unique peptides. Error bars indicate standard deviation of technical triplicates.

**Figure 2 ijms-22-12999-f002:**
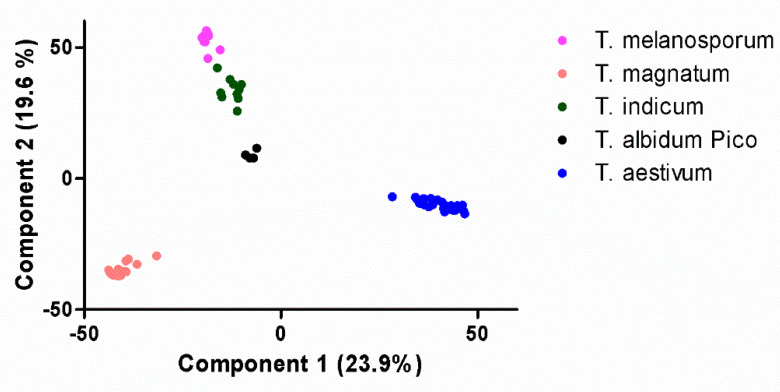
Principal component analysis (PCA) sample projection of the first two components for truffle species. The PCA was performed on 2715 proteins that were identified in all samples.

**Figure 3 ijms-22-12999-f003:**
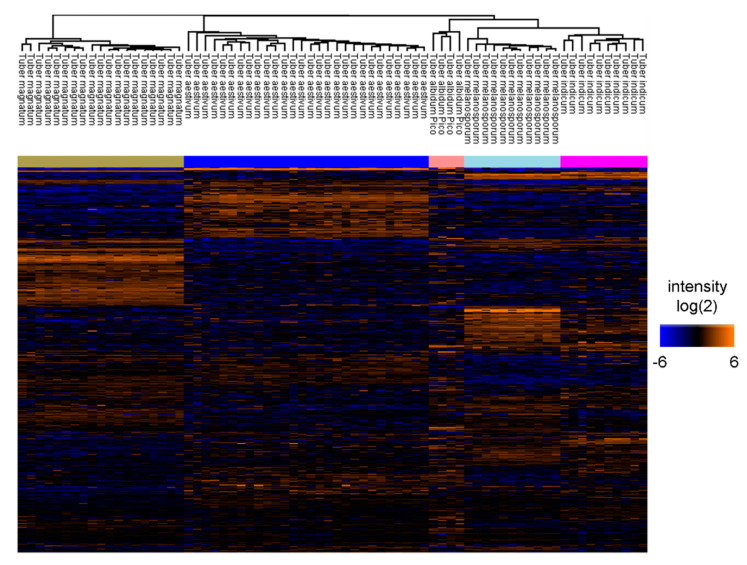
Heat map using ANOVA [21] test with 5% FDR displaying significant protein abundance in different truffle species after hierarchical clustering. The quantities of the different proteins of the truffle species were compared against each other. The heat map was generated by Perseus and displays log2 protein areas for the 2066 ANOVA significant proteins identified in all samples. Each column in the heat map represents a different sample. Each line represents a protein. Orange lines correspond to proteins with a high abundance within the comparison of the five truffle species, blue lines with proteins of a low abundance.

**Figure 4 ijms-22-12999-f004:**
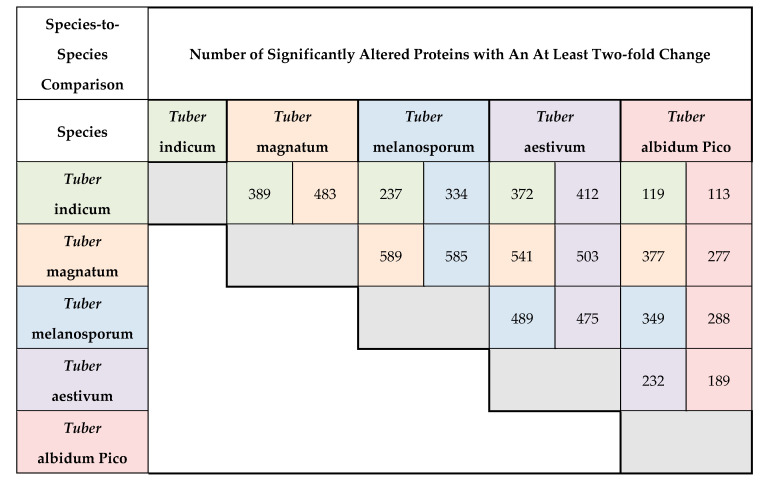
Results of the species-to-species comparison listing the number of 1% FDR T-test significant proteins upregulated with an at least two-fold change. Colors correspond to truffle species. Comparisons of species are done horizontally versus vertically. The number of significantly upregulated proteins with an at least two-fold change is listed for each species-against-species comparison. As there are two numbers resulting from each comparison (upregulated in species horizontally listed and upregulated in species vertically listed), the numbers are color coded accordingly to the species they belong to.

**Table 1 ijms-22-12999-t001:** Top 10 hits of enriched biological processes in ANOVA significant proteins from the comparison of all truffles.

GO ID	GO Description	*p*-Value	Corrected *p*-Value	Cluster Frequency	Total Frequency
44281	small molecule metabolic process	5.66 × 10^−70^	9.92 × 10^−67^	368/1325 (27.7%)	1195/10,365 (11.5%)
6082	organic acid metabolic process	1.27 × 10^−53^	1.11 × 10^−50^	243/1325 (18.3%)	710/10,365 (6.8%)
19752	carboxylic acid metabolic process	6.71 × 10^−51^	3.92 × 10^−48^	232/1325 (17.5%)	679/10,365 (6.5%)
43436	oxoacid metabolic process	4.08 × 10^−49^	1.79 × 10^−46^	232/1325 (17.5%)	693/10,365 (6.6%)
55114	oxidation–reduction process	2.36 × 10^−40^	8.26 × 10^−38^	353/1325 (26.6%)	1438/10,365 (13.8%)
6520	cellular amino acid metabolic process	2.08 × 10^−35^	6.08 × 10^−33^	164/1325 (12.3%)	481/10,365 (4.6%)
1901605	alpha−amino acid metabolic process	8.26 × 10^−32^	2.07 × 10^−29^	114/1325 (8.6%)	285/10,365 (2.7%)
8152	metabolic process	4.30 × 10^−30^	9.42 × 10^−28^	1100/1325 (83.0%)	7285/10,365 (70.2%)
6091	generation of precursor metabolites and energy	2.46 × 10^−29^	4.80 × 10^−27^	97/1325 (7.3%)	230/10,365 (2.2%)
1901564	organonitrogen compound metabolic process	8.41 × 10^−29^	1.48 × 10^−26^	604/1325 (45.5%)	3314/10365 (31.9%)

**Table 2 ijms-22-12999-t002:** Top 15 hits of enriched biological processes in upregulated proteins from *T. magnatum* in comparison with *T. indicum*.

GO ID	GO Description	*p*-Value	Corrected *p*-Value	Cluster Frequency	Total Frequency
44281	small molecule metabolic process	4.71 × 10^−17^	4.02× 10^−14^	93/331 (28.0%)	1195/10,365 (11.5%)
55114	oxidation–reduction process	9.55 × 10^−14^	4.08 × 10^−11^	97/331 (29.3%)	1438/10,365 (13.8%)
8152	metabolic process	2.72 × 10^−13^	7.73 × 10^−11^	288/331 (87.0%)	7285/10,365 (70.2%)
6082	organic acid metabolic process	5.80 × 10^−13^	1.24 × 10^−10^	61/331 (18.4%)	710/10,365 (6.8%)
19752	carboxylic acid metabolic process	3.06 × 10^−12^	5.22 × 10^−10^	58/331 (17.5%)	679/10,365 (6.5%)
96	sulfur amino acid metabolic process	3.72 × 10^−12^	5.30 × 10^−10^	17/331 (5.1%)	61/10,365 (0.5%)
1901605	alpha-amino acid metabolic process	5.37 × 10^−12^	6.55 × 10^−10^	35/331 (10.5%)	285/10,365 (2.7%)
43436	oxoacid metabolic process	7.12 × 10^−12^	7.60 × 10^−10^	58/331 (17.5%)	693/10,365 (6.6%)
9069	serine family amino acid metabolic process	5.88 × 10^−11^	5.58 × 10^−09^	16/331 (4.8%)	62/10,365 (0.5%)
6091	generation of precursor metabolites and energy	2.09 × 10^−10^	1.78 × 10^−08^	29/331 (8.7%)	230/10,365 (2.2%)
6534	cysteine metabolic process	2.11 × 10^−09^	1.63 × 10^−07^	11/331 (3.3%)	32/10,365 (0.3%)
6520	cellular amino acid metabolic process	2.29 × 10^−09^	1.63 × 10^−07^	42/331 (12.6%)	481/10,365 (4.6%)
6790	sulfur compound metabolic process	1.16 × 10^−08^	7.60 × 10^−07^	25/331 (7.5%)	209/10,365 (2.0%)
1901564	organonitrogen compound metabolic process	1.42 × 10^−08^	8.66 × 10^−07^	154/331 (46.5%)	3314/10,365 (31.9%)
70813	hydrogen sulfide metabolic process	2.20 × 10^−08^	1.17 × 10^−06^	7/331 (2.1%)	12/10,365 (0.1%)

## Data Availability

The mass spectrometry proteomics data have been deposited to the ProteomeXchange Consortium via the PRIDE partner repository [38] with the dataset identifier PXD027871.

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
