# Peer review of "Investigation of the Proteomes of the Truffles Tuber albidum pico, T. aestivum, T. indicum, T. magnatum, and T. melanosporum"

_ijms, 2021, doi:10.3390/ijms222312999_

Round 1

Reviewer 1 Report

In their manuscript titled ""Investigation of the proteomes of the truffles Tuber albidum pico, T. aestivum, T. indicum, T. magnatum, and T. melanosporum" Krosser D. and colleagues, by making use of differential quantitative bottom-up proteomics approach, analysed five different truffles species, including the delicious and noble T. magnatum, which deserves an international fair that is held in Alba (ITALY) just during these days . The volatile organic compounds (VOCs) of the white truffle (T. magnatum) have been matter of investigation because of the intense aroma exhaled by the fruiting body. However, up to date the proteome of truffles has been scarcely explored. By data-independent acquisition (DIA) proteomics mode, the authors managed to identify and quantify 2,715 different proteins that were present in all truffles analysed. Quantitative proteome fingerprints were generated and gene ontology enrichment analysis was performed. On the whole the study highlights that the superior aroma of T. magnatum and T. melanosporum is associated with the enrichment of proteins involved in sulfur-metabolic process.   Prior publication the authors need to clarify a couple of issues that are quickly below discussed:
  1. Introduction needs to be slightly revised, because references not always fit within the context. For example, lines 43-47. Vita et al. investigated the volatile organic compounds and in that paper (Scien. Rep.) there is no record/trace of 2D-gel electrophoresis since the authors focused the inquiry to VOCs. Please check also throughout the main text;
  2. When it concerns the statistical comparison of the protein abundance among the different truffles species in the figure 3 legend it is not clear what has been used as control. The authors should clarify this question;
  3. There is no Table 4. I guess that it is simply a typo and that the authors meant Figure 4;
  4. Figure 4 is not so clear how the species-to-species comparison has been carried-out, thus it is not so simple and explicit to read the table. Please describe the results more extensively and in detail in the "Results" section and then in the figure caption;
  5. Based on the previous work done by Vita et al. it appears that, besides sulfur-containing compounds, truffles VOCs include aldehydes, alcohols, esters, ketones, aromatic compounds, terpenes and other unidentified molecules. Could the authors discuss a little bit accurately whether among the proteins that are classified according to the Gene Ontology into the classes of "oxidation-reduction process", "carboxylic acid metabolic process", "small molecules metabolic process", there are protein members/enzymes involved in the biosynthesis, or alternatively in the degradation, of VOCs other than sulfur-containing compounds.

Author Response

Dear reviewer,

thank you very much for your questions and comments.

Please find attached our response to your concerns.

Best regards,

Hartmut Schlüter

Reviewer 2 Report

Krösser et al. generated proteomic data for three truffle species, and used data from two other species (I think), to understand the make-up of the proteome, and understand how it varied between species. The methodologies looked fine to me, however, someone with more experience in proteomic data generation would be more qualified to judge this. The largest flaw of this study is that it appeared to be lacking a clear motivation. I expand on this below in the broad comment sections. I think this is really neat work, but I am left wondering why did you do this, and how does it fit into the body of work already generated on truffles and their volatome. I would spend time really working this into a method appropriate for detecting food fraud, or for a biological study on how conserved the truffle proteomes are, and with that, I think it would be a really nice piece of work. A smaller detail as well: be sure to be explicit in what data you generated in this study. It was really hard to figure out (and I still haven’t), where the data from the other two species came from, and how they fit into this story.  

Broad comments:

Title: Why does the title include five truffle species when only three were analyzed in this study? Where is the data coming from from the other two species (because they’re included in the figure)? During the whole paper it looks like you only worked on three species but there are data for five, so more clarity on what you generated here is needed. Using previously generated data is good, but it just needs to be stated somewhere

Abstract: You should state the names of the species you analyzed—give more detail in that regard. Also, again, this might need to be restructured based on how you decide to focus this study.

Introduction: I think the introduction can be more clear about your motivations for this study. Instead of investigating the proteomes of three different species for fun, why did you really do this? That hasn’t come through yet. You hint around the idea of using proteomics to help against food fraud. I would dive more into that. What is food fraud? What does food fraud in truffles look like, and how can this methodology help with that? Or perhaps were their other motivations for the study? You can include that. I think it’s fine to explain the methodologies, but the questions, in my mind are more important. Also,  think you need to include the three species you analyze somewhere in here, because that’s missing (and also from the abstract).

Discussion: Again, I think you need to spend more time discussing the importance of your findings to the main thrust of this paper, which could be food fraud if developed that way. I would spend more time delving into the specifics of this and its importance. Also, I would devote more space to putting your results in the context of what is known about truffles and their volatiles. There is a significant body of work studying those, so how does your work specifically fit into that. You have a neat biological story here, and a neat food fraud method here, so I would spend some time trying to refocus this work to focus on that!

I like Figure 2 (the PCA). This leaves me wondering though, are T. melanosporum, T. indicum and T. albidum Pico evolutionarily more closely related? There is plenty of work on this subject, and I think that would be a nice addition, because then another of your findings could be that that the expressed proteome of truffles is more similar if they are closely related (or the inverse would be fascinating too… if they aren’t more closely related to one another, and give plausible explanations for why this is the case). But I think that is something that needs to be addressed. Looking at the paper: Historical biogeography and diversification of truffles in the Tuberaceae and their newly identified southern hemisphere sister… , I can see that T. melanosporum and T. indicum are sister species. So I think it would be worthwhile to explore these results from an evolutionary perspective as well.

Smaller comments:

L12, L31: Tuber should be italicized

L14: Sentence here doesn’t make any sense.

L31: First sentence could be rewritten as: Truffles are ascomycete fungi in the genus Tuber that form subterranean fruiting bodies.

L39: remove “responsible” or keep it and put “distinct aroma” after

L42: include “by” before “analyzing intact proteins”

L42-43: include that it was the genome that was sequenced

L49-50: Fix the sentence here (not sure it’s a whole sentence)

L57: I would say “benefitted” instead of “profited”

L64-67: I think this sentence could be reworded here to make it more approachable to readers simply interested in truffles who are not well-versed in proteomics

L77-78: Why did you test different sample preparation protocols? I would just include a brief statement in this opening sentence along the lines of saying that it was necessary to determine which recovered the most proteins for ground truffle powder.

L117-118: Where did the data from the other two species come from???

L270: You must include the names of the three species here, and how many replicates per species were included? Also, I know truffle aroma varies by maturity of the fruiting body, so were they all at the same maturity? More description is needed here, especially since this could bias results.

L309: include “for” before 10 min; check this whole paragraph over for missing prepositions

L426: changed “allowed” to “allow”

L459-460: this isn’t a complete sentence.

Figures:

Figure 1: Somewhere in the legend this should say what species was tested? Were all species tested and this is the average?

Also, I would changed the font here because it’s all bleeding together and really hard to read. Also, I am not sure that you need patterns in the bars because the labels are sufficient.

Figure 3: I would change the color of this figure to blue and yellow because red and green can be difficult to disentangle for colorblind people.

Author Response

(The authors gave the same response as above.)
